# Effects of Thyroid Hormone on Tissue Hypoxia: Relevance to Sepsis Therapy

**DOI:** 10.3390/jcm10245855

**Published:** 2021-12-14

**Authors:** Athanasios I. Lourbopoulos, Iordanis S. Mourouzis, Athanasios G. Trikas, Ioulia K. Tseti, Constantinos I. Pantos

**Affiliations:** 1Department of Pharmacology, National and Kapodistrian University of Athens, 75 Mikras Asias Ave, Goudi, 11527 Athens, Greece; alourbop@med.uoa.gr (A.I.L.); imour@med.uoa.gr (I.S.M.); cgaki@med.uoa.gr (A.G.T.); dep-pharmacology@med.uoa.gr (I.K.T.); 2Institute for Stroke and Dementia Research (ISD), University of Munich Medical Center, 81377 Munich, Germany; 3Neurointensive Care Unit, Schoen Klinik Bad Aibling, Kolbermoorerstrasse 72, 83043 Bad Aibling, Germany

**Keywords:** thyroid hormone, sepsis, hypoxia, microcirculatory failure, HIF-1α

## Abstract

Tissue hypoxia occurs in various conditions such as myocardial or brain ischemia and infarction, sepsis, and trauma, and induces cellular damage and tissue remodeling with recapitulation of fetal-like reprogramming, which eventually results in organ failure. Analogies seem to exist between the damaged hypoxic and developing organs, indicating that a regulatory network which drives embryonic organ development may control aspects of heart (or tissue) repair. In this context, thyroid hormone (TH), which is a critical regulator of organ maturation, physiologic angiogenesis, and mitochondrial biogenesis during fetal development, may be of important physiological relevance upon stress (hypoxia)-induced fetal reprogramming. TH signaling has been implicated in hypoxic tissue remodeling after myocardial infarction and T3 prevents remodeling of the postinfarcted heart. Similarly, preliminary experimental evidence suggests that T3 can prevent early tissue hypoxia during sepsis with important physiological consequences. Thus, based on common pathways between different paradigms, we propose a possible role of TH in tissue hypoxia after sepsis with the potential to reduce secondary organ failure.

## 1. Introduction

It is well recognized that thyroid hormone (TH) metabolism is altered under several conditions in which hypoxia prevails, such as myocardial infarction, stroke, trauma, or sepsis. All these conditions induce low circulating L-triiodothyronine (T3) levels, known as non-thyroidal syndrome (NTIS), which is directly associated with their prognosis [1,2,3,4]. At the same time, there is a growing amount of experimental evidence that TH signaling is implicated in the pathophysiology of hypoxic-induced tissue damage and remodeling, while TH treatment can positively affect both. Sepsis implicates from the very beginning in its cascade of events a microcirculatory failure with tissue hypoxia which eventually leads to tissue and organ damage, for which no effective treatment exists thus far [5,6].

Βased on the above new evidence, we reviewed the role of thyroid hormone in tissue hypoxia and particularly in the setting of sepsis. We conclude that this may be of therapeutic importance since sepsis carries a high morbidity and mortality despite current treatments.

## 2. THs and Tissue Hypoxia

Tissue hypoxia is the pathophysiological condition resulting from the imbalance between cellular O2 consumption and vascular O2 supply in the tissue. This induces a relative lack—but not complete absence (i.e., anoxia)—of O2 in eukaryotic cells and activation of the conserved “hypoxia response pathway”. The pathway begins with the reversible stabilization of the hypoxia-inducible factor alpha (HIF-1α), continues with its dimerization with HIF-1β, and ends to stimulation of 500–1000 target-genes [7,8,9]. The results are increase in metabolism, angiogenesis, erythropoiesis, cell migration and invasion, cell proliferation, and inflammation (for detailed reviews see [8,9]).

Evidence suggests that TH interacts with HIF-α and the hypoxia response pathway. Thus, L-thyroxine (T4) and T3 can induce HIF-1α activity by both genomic and nongenomic mechanisms in cancer cells [10]. T3 can nongenetically regulate expression of HIF-1α via activation of (phosphatidylinositol 3-kinase) (PI3K) and mitogen-activated protein kinases (MAPK-) signaling cascades. Thyroid hormone receptors (TRα and TRβ) are implicated in these actions (see also Section 4.2) [11]. Similarly, T3 mediates increased synthesis of HIF-1α on primary human hepatic cells as well [12]. On the other hand, HIF-1 reduces cellular TH signaling upon tissue hypoxia through induction of Deiodinase3 (DIO3), as a protective mechanism to reduce metabolic rate upon lack of O2 [13]. Based on the above, TH availability could be considered as an important determinant of the hypoxia response.

Accumulating experimental evidence shows that TH plays a critical role in the response of various tissues to ischemic/hypoxic injury, as shown in Table 1 and Table 2. We were the first to show that in an experimental model of ischemia–reperfusion using isolated rat hearts, T4 pretreatment improved post-ischemic recovery in a similar pattern as ischemic preconditioning. Interestingly, both T4 pretreatment and ischemic preconditioning were shown to suppress the ischemia–reperfusion (I/R)-induced activation of the pro-apoptotic p38MAPK [14,15]. Along this line, TH was also shown to upregulate cardioprotective molecules such as heat-shock protein 27 (HSP27) and heat-shock protein 70 (HSP70) which were also involved in the underlying mechanisms of ischemic preconditioning [16,17]. In accordance, acute T3 administration at reperfusion (at a high dose which had no effect on non-ischemic myocardium) enhances post-ischemic recovery of function and reduces myocardial injury, as indicated by apoptosis and tissue necrosis markers, via inhibition of the p38MAPK activation [18,19]. In this experimental setting, T4 does not have a protective effect. A dose-dependent effect of T3 on cardioprotection against ischemia–reperfusion injury in isolated rat hearts has also been shown by preserving calcium-cycling proteins [20]. In vivo studies of cardiac ischemia–reperfusion have shown that T3 prevents mitochondrial impairment and cell loss [21]. Interestingly, in a very recent report, development of T3 polymeric nanoparticles for targeted delivery was shown to protect neonatal cardiomyocytes against hypoxic damage [22]. Furthermore, TH prevents cardiac remodeling in experimental acute myocardial infarction [23] and in patients with reperfused myocardial infarction.

TH-induced protection against injury has been demonstrated in other organs as well, suggesting a conserved mechanism [26,27]. As shown in Table 2, thyroid hormones were found to induce cytoprotection against ischemic injury in the kidney [28,29]. T4 administration immediately or 24 h after kidney ischemia resulted in higher inulin clearance and preserved cellular integrity [29]. Similarly, T3 administration 24 h prior to renal ischemia could precondition against ischemia–reperfusion (I/R) injury. This was evident by a marked decrease in I/R-induced proteinuria, improved lipid peroxidation biomarkers, and increased antioxidant enzymes [28]. Pre-treatment with T4 was also found to preserve cellular structure in a model of anoxia/reoxygenation injury in isolated rabbit proximal tubule cells [30]. In addition, a single dose of T3 improved the clinical signs and acute tubular necrosis after renal I/R injury [31]. In accordance, preconditioning with T3 has been reported to protect against renal I/R injury by inhibiting apoptosis of tubular epithelial cells [32]. In the liver, pretreatment with a single dose of T3 in rats significantly diminished hepatocellular injury induced by ischemia–reperfusion (I/R) [33]. Yang et al. also provided evidence indicating that antecedent T3 injection preserves liver function, and reduces histological damage and apoptosis after I/R by enhancing autophagy [34]. Furthermore, T_3_ is shown to trigger protection of the liver against I/R injury by suppressing the inflammatory response and IL-1β upregulation with AMPK playing a causal role [35]. TH seems to be crucial in the response to lung injury in experimental sepsis and ventilator-induced trauma [36]. Furthermore, TH increases alveolar fluid clearance in normal and hyperoxia-injured lungs [37].

In the brain, the effects of TH on hypoxia are not fully understood. Various models of brain hypoxia indicate that TH ameliorates hypoxic injury, but results are contradicting, suggesting more complex mechanisms. T4 treatment in rat pups reversed the hypoxia-induced white matter injury and death of pre-oligodendrocytes by upregulation of brain-derived neurotrophic factor-TrkB (BDNF) signaling in the immature brain in both cortex and white matter [38,39]. In cultured mouse primary neurons, T3 reduced hypoxia-induced neuronal damage and apoptosis. [40]. On the other side, ischemia/hypoxia induces an acute (within 1 h) translocation of Dio3 to the neuronal nucleus, facilitating T3 reduction proximal to the TH-receptors and thus decreased TH signaling. This causes a state of cell-specific hypothyroidism which reduces cellular metabolism to reduce acute ischemia-induced hypoxic neuronal damage [41]. Interestingly, experimental chronic hyperthyroidism or thyrotoxicosis in rats aggravated post-stroke ischemic injury induced by middle cerebral artery occlusion [42,43]. These data cumulatively suggest direct effects of TH on the brain, but different responses according to the levels of supplied O2 (hypoxia, anoxia or ischemia).

**Table 2 jcm-10-05855-t002:** Accumulating experimental evidence shows that thyroid hormone plays a critical role in adaptation of the kidney, liver, lungs, and brain to hypoxic/ischemic injury (I/R: Ischemia/Reperfusion, MCAO: Middle Cerebral Artery Occlusion, HI: Hypoxic Injury).

Study	Type of Treatment	Model	Outcome
Ferreyra et al., 2009 [28]	Pre-treatment with T3	In vivo I/R in rat kidney	Reduced proteinuria
Erkan et al., 2003 [30]	Pre-treatment with T4	Anoxia-reoxygenation in rabbit proximal tubule cells	Better preservation of cellular structure
Sutter et al., 1988 [29]	Treatment with T4 post-ischemia	In vivo I/R in rat kidney	Improved kidney function, preserved cellular morphology
Ferreyra et al., 2013 [31]	Pre-treatment with T3	In vivo I/R in rat kidney	Improved clinical signs and acute tubular necrosis
Kim et al., 2014 [32]	Pre-treatment with T3	In vivo I/R in rat kidney	Protection of tubular epithelial cells against apoptosis
Fernandez et al., 2007 [33]	Pre-treatment with T3	In vivo I/R in rat liver	Reduced injury (serum AST and ALT levels)
Yang et al., 2015 [34]	Pre-treatment with T3	In vivo I/R in rat liver	Improved liver function, reduced histological damage and apoptosis
Vargas and Videla 2017 [35]	Pre-treatment with T3	In vivo I/R in rat liver	Reduced liver injury
Bhargava et al., 2008 [37]	Pre-treatment with T3	Hyperoxia injury in rat lung	Increased alveolar fluid clearance
Hiroi et al., 2006 [44]	Treatment with T4 post-ischemia	Transient focal ischemia in mouse brain	Reduced cerebral infarct volume, and improved neurological deficit score
Hung et al., 2013 [39]	T4 treatment immediately after HI	Hypoxia in immature rat brain	Protected against white matter injury
Hung et al., 2018 [38]	T4 treatment after HI	Right carotid-artery ligation, followed by hypoxia	Protected against white matter injury
Li et al., 2019 [40]	T3 treatment during hypoxia	Mouse primary cortical neurons	Reduced neuronal damage
Keshavarz et al., 2017 [42]	Chronic T4 pre-treatment	MCAO in rats	Enhanced injury
Rastogi et al., 2008 [43]	Chronic T4 pre-treatment	MCAO in rats	Enhanced injury

## 3. TH and Angiogenesis

THs serve a critical role in angiogenesis during development [45,46,47] and can regulate microcirculation, switch pathologic to physiologic angiogenesis, and regulate red blood cell rheology. A significant amount of research is also focused on TH’s effects in relation to pathological angiogenesis by several types of cancers [48], but this exceeds the scope of the present work. The molecular mechanisms of the angiogenetic action of THs are both non-genomic and genomic, as previously reviewed in detail [47]. Angiogenic genes such as VEGF-A, VEGF, HIF2a, and angiopoietin-2 (ANG2) are found under direct control of THs (more noticeable with T4 than T3) [49]. THs (mainly T4) are directly pro-angiogenic via alpha V beta 3 (αvβ3) integrin receptors on the plasma membrane and a MAPK-dependent and/or fibroblast growth factor (FGF2) downstream effect [50,51,52]. Due to the above, THs could potentially affect and modulate the results of tissue hypoxia that lead to neo-vessel formation with abnormal vasomotor response, increased vascular permeability, and thrombosis (pathologic angiogenesis), with further aggravation of hypoxia (hypoxia vicious circle) and organ failure [6,46].

The potential induction of angiogenesis by T3 has been investigated in the adult heart from hypothyroid mice [53]. Hypothyroidism resulted in cardiac microvascular impairment and rarefaction with increased sensitivity to angiogenic growth factors. Treatment with T3 induced cardiac sprouting angiogenesis in adult hypothyroid mice via a significant increase in platelet-derived growth factor receptor beta (PDGFR-b) protein levels in the hypothyroid heart. The fact that this effect of T3 is PDGFR-b mediated was proven when PDGFR inhibitors blocked the action of T3 both on sprouting angiogenesis in left ventricular tissue and on capillary growth in vivo. In addition, activation of Akt-signaling mediated the T3-induced angiogenesis and was blocked by PDGFR inhibitor and neutralizing antibody [53]. Similar effects are also found in other organs, suggesting a conserved role of TH. For example, THs stimulate brain angiogenesis in vivo and in vitro via vascular endothelial growth factor A (VEGF-A) and basic fibroblast growth factor 2 (FGF2) in healthy [54] or hypothyroid rats. In the liver, T3 seems to stimulate regeneration after partial hepatectomy and at least partially via a VEGF-mediated increase in angiogenesis [55]. In human umbilical vein endothelial cells, T4 can increase the expression of bFGF mRNA via the non-genomic integrin-αvβ3/PKD/HDAC5 signaling pathway and promote angiogenesis in vitro [56]. Collectively, the availability of TH at the tissue level during hypoxia may regulate the angiogenetic response.

## 4. Sepsis and Tissue Hypoxia

Sepsis is a complex, life-threatening, disorder that develops as a dysregulated host response to an infection and is associated with acute organ dysfunction and a high risk of death [57]. Annually, 2.8 million deaths are attributable to sepsis [58]. The world health assembly and WHO made sepsis a global health priority in 2017 and have adopted a resolution to improve the prevention, diagnosis, and management of sepsis [58]. Sepsis is mainly bacterial-induced (bacterial sepsis), but in the cases of culture-negative sepsis a viral cause (viral sepsis) can also be considered, especially in vulnerable patients [59,60] or within the spectrum of a COVID-19 infection [61]. Despite current treatments, overall mortality of sepsis is 25% [62] and exceeds 50% for intrahospital sepsis with (multi-) organ failure (MOF) [63].

According to recent guidelines (“surviving sepsis campaign 2016”), treatment of sepsis should begin within the first (“golden”) hour after diagnosis [62]: the relevant microbiological samples should be collected and appropriate, empiric, broad-spectrum, antibiotics must be immediately administered intravenously. In the case of septic shock, stabilization of the organ-perfusing blood pressure (macro-circulation) must be achieved within the first hour through the administration of crystalloid solutions and, if necessary, the use of inotropic-vasoactive agents (noradrenaline and vasopressin, dopamine, dobutamine, or adrenaline upon relevant indications); administration of hydrocortisone may also be considered [62].

However, despite the stabilization of macro-circulation and the restoration of oxygen levels in blood, sepsis often results in primary and secondary MOF due to microvascular disorders and tissue hypoxia at the cellular level (reviewed in detail [6,58,62,64]). Specifically, 40–50% of septic patients develop renal failure [65], 35% hepatic failure [66], 6–9% secondary respiratory failure [67,68], and 34% develop secondary leukopenia and immunosuppression [69], while the rates of secondary cardiac, cerebral, and gastrointestinal disorders may vary. Secondary damage of the coagulation system, which is responsible for increased rates of thrombosis, or even disseminated intravascular coagulation, is also evident [70]. On the other hand, emerging evidence suggests that even the widely recommended inotropes and vasoactives for macrocirculatory support during treatment of sepsis [62] increase mortality of patients with septic shock. In a recently published series of 417 patients with septic shock, the use of epinephrine and dobutamine was surprisingly associated with significantly higher in-hospital mortality [71], particularly in patients with severe ischemia and high lactate levels due to septic shock [71].The potential mechanisms of these unfavorable effects of certain agents, which are commonly used to support hemodynamics in the critically ill patient, are not well understood. Eventually, accumulation of lactate, despite the restoration of macro-circulation, is highly associated with mortality, has a high prognostic value, and is a direct indicator for both tissue ischemia at the cellular level and micro-circulatory damage [72].

### 4.1. Microcirculatory Dysfunction in Sepsis Leads to Tissue Hypoxia

The precise mechanisms of sepsis-induced cell injury and organ dysfunction are not fully understood and continue to be an active area of scientific investigation (for detailed reviews please refer to [6,58,62,64]). Nevertheless, it appears that tissue hypoxia occurs because of either systemic or local mismatch between oxygen delivery and tissue demand. Interestingly, even after aggressive hemodynamic stabilization of the septic patient, a normal or high cardiac output is typically achieved, yet tissue perfusion can remain markedly impaired due to reduced microvascular perfusion [5]. The precise mechanisms involved are not fully elucidated but include a reduction in the number of perfused capillaries due to several factors such as changes in erythrocytes’ deformability and aggregation, endothelial cell dysfunction with increased permeability and apoptosis, altered vasomotor tone, increased numbers of activated neutrophils and neutrophil–endothelial interactions due to over-expression of endothelial surface adhesion molecules, and activation of the clotting cascade [6,73,74]. Preclinically, a normodynamic model of sepsis with cecal ligation and perforation (CLP) in rats (CLP model), shows a 36% reduction in perfused capillary density and a 265% increase in the “non-perfused” capillaries in striated muscle [75], indicating the development of microvascular failure at the tissue level. A similar failure is also reported within other experimental settings [76,77], suggesting a common tissue reaction to stress. Furthermore, gut and muscle blood flow heterogeneity were shown to increase, resulting in impaired oxygen extraction after endotoxin administration or fecal peritonitis [78,79]. Additionally, erythrocytic abnormalities and aggregation occur early in sepsis and can impair hemorheology with impact on microcirculation and tissue perfusion [80]. In agreement to this experimental evidence, changes in capillary perfusion have been demonstrated in patients with sepsis and the severity of perfusion defect is associated with poor outcome [64]. Taken together, sepsis-induced hypoxia is mainly a disorder at the level of microcirculation.

Clinical studies show that tissue hypoxia, which is measured as an insufficient lactate clearance in blood during the first hours of a patient’s recovery, is associated with MOF and increased mortality [81]. Another study showed that there is a strong correlation between the improvement of lactate levels in the first 6 h and a consequent improvement of blood biomarkers over 72 h as well as improvement of multi-organ dysfunction [82]. Increased lactate clearance is a bio-marker strongly associated with improved micro-circulation and resulting tissue hypoxia. On the other hand, a reduced lactate clearance in septic patients is associated with increased caspase-3 at 72 h, indicating increased tissue apoptosis as a result of tissue hypoxia [82]. Cumulatively, these data suggest that tissue hypoxia is a primary factor with a critical role in the pathophysiology of MOF in sepsis and not an end-stage phenomenon.

### 4.2. The Molecular Basis of Tissue Hypoxia in Sepsis: HIF-1a and p38 MAPK

Molecularly and mechanistically, sepsis is a complex, inappropriately amplified process triggered by pathogen and host components (for a detailed review of these mechanisms, please refer to [83]). As a common path, the activated inflammatory cascade and the released pathogen-associated molecular patterns (PAMPs) and danger-associated molecular patterns (DAMPs) pose a strong stress on tissues and endothelium via surface receptors and intracellular mediator kinases [83]. Within the paradigm of sepsis, microcirculatory failure occurs when endothelium mechanically and molecularly is affected and compromised via these processes [6,83]. The resulting tissue hypoxia activates a fetal-like cellular re-programming in sepsis and results in tissue remodeling and organ failure [84,85].

The activation of HIF-1α is considered as one of the primary molecular mechanisms responsible for this response and is a common downstream signal in every tissue-hypoxia case [8,86]. A threshold of oxygen tissue content seems to exist for HIF-1α signaling: oxygen partial-pressure (pO2) levels in tissue below 10mmHg lead to activation of HIF-α dependent regulatory mechanisms [87]. Hypoxia-independently, HIF-1 can also be activated by metabolic signaling pathways or growth factor-signaling pathways such as MAPK- and PI3K- signaling cascades [10], capable of creating–under conditions–closed loops with the latter systems (see below). HIF-1α activity increases phagocytes’ survival and stimulates the expression of proinflammatory cytokines such as tumor-necrosis factor (TNF), interleukins-1 and -12 (IL-1, IL-12), and angiogenic factors such as the vascular endothelial growth factor (VEGF) [86]. VEGF is essential for angiogenesis and a potent inducer of vascular permeability via p38MAPK activation (pathologic angiogenesis) [88]. Inhibition of p38MAPK improves lung permeability, attenuates inflammation, and alleviates lung injury in experimental sepsis [89]. In an experimental paradigm of infection, HIF-1α activation in alveolar epithelial cells resulted in dysfunctional alveolar remodeling. Moreover, the hypoxia-dependent HIF-1α activity appeared to determine epithelial cell fate, as mouse epithelial cells without HIF-1α expression have been recovered more rapidly with improved expansion of the type II alveolar cell population [90]. Similarly, HIF-1α activation and proinflammatory response are also elicited in COVID-19 patients and infected cells [91]. HIF-1α also triggers in sepsis a metabolic switch towards glycolysis (fetal phenotype) with important physiological consequences [92] and additional promotion of M1- (proinflammatory) macrophage activity and response [93]. Taken together, hypoxia-induced HIF-1α activation is an additional “second key-trigger” for tissue remodeling after sepsis (Figure 1).

Following hypoxia and HIF1-α upregulation, p38MAPK is a “second-stage amplifier” triggered by the presence of tissue hypoxia and amplified by HIF-1α [94]. The p38 MAPK protein kinases are a class of evolutionarily conserved molecules that link external stimuli (such as PAMPs, cytokines, osmotic stress) with downstream effectors through phosphorylation of proteins and/or lipids [95,96]. p38 MAPK is a key regulator for inflammation, cell-cycle regulation, cell death, development, differentiation, and senescence [95]. p38 MAPK mediates its actions via phosphorylation of a broad range of substrates (reviewed in detail in [96]). In brief, phosphorylation of MSK (mitogen- and stress-activated kinase) 1 and 2 can regulate transcription factors such as c-AMP Response Element-Binding Protein (CREB), c-AMP-dependent transcription factor (ATF-1), the NF-κB isoform p65, signal transducer and activator of transcription (STAT 1 and 3), and p53. Notably, p38 MAPK can phosphorylate Hsp27 (heat-shock protein 27), mediating actin filament remodeling. A large number of cytosolic proteins important for cell proliferation and survival/apoptosis are shown to be regulated by p38 MAPKs, including phospholipase A2, the tau protein, cyclin D1, and Bcl-2 family proteins [96]. Another example of p38 MAPK substrate is the FGFR1 (fibroblast growth factor receptor 1), which is translocated from the extracellular space into the cytosol and nucleus, regulating cell growth. Furthermore, p38 MAPK is directly implicated in transcription of several genes involved in the inflammatory response, such as IL-6, IL-8, IL-12p40, and MCP-1 (monocyte chemoattractant protein 1) via histone H3 phosphorylation [96]. Upon stress, p38 MAPK is activated and results either in cell apoptosis, further promotion of LPS-induced cytokine production during the inflammation, or eventually cellular proliferation/differentiation and tissue remodeling [96]. Tissue hypoxia induces a sustained activation of p38 MAPK and tissue injury while interventions that suppress p38 MAPK activity increase cell survival [19,97]. This indicates a constant deadly interplay between p38 MAPK and hypoxia. Interestingly, viruses and hypoxia share common cellular kinase signaling pathways which on the one hand facilitate viral entry and replication, and on the other hand promote cellular apoptosis [98]. For example, influenza-A can also replicate via p38 MAPK activation, the same pathway which leads to cellular apoptosis [98,99]. Hypoxia can enhance tissue injury by increasing viral load in cells, a mechanism which has not been appreciated. Here, inhibition of p38 MAPK is shown to diminish viral replication, viral ribonucleoprotein export, and apoptosis. Influenza virus, like hypoxia, induces perturbations of the intracellular redox balance, resulting in increased production of reactive oxygen species (ROS), which can also activate p38 MAPK. Additionally, NADPH oxidase 4 (NOX 4)-regulated p38MAPK activation can result in increased ROS production [100]. Along with this line, in an experimental model of viral myocarditis, Bosentan, an endothelin-1 receptor (ET1R) antagonist, although improving cardiac function, enhanced viral load and myocarditis severity due to p38 MAPK activation [101]. In contrast, administration of SB203580, an inhibitor of p38 MAPK, attenuated viral replication in the heart, myocardial damage, and preserved cardiac function [101]. Collectively, p38 MAPK is a key regulator of cellular reactions and survival within the complex cascade of sepsis.

Interestingly, experimental data support that the activation of p38 MAPK in response to hypoxia is further enhanced by certain vasoactives and inotropes, aggravating tissue injury. Similarly, we have shown that sympathomimetic agents aggravated ischemia via p38MAPK in models of myocardial ischemia–reperfusion [102]. On the other hand, pharmacological inhibition of p38MAPK via dexmedetomidine protected the intestine and kidneys from ischemia/reperfusion-related mitochondrial apoptosis and inflammatory response [103,104]. Thus, appropriate pharmacological modulation of the p38MAPK pathway could diminish hypoxia-induced tissue injury in sepsis [105] (Figure 1).

## 5. Sepsis and Neurohormonal Response

Sepsis imposes severe stress on the body and leads to an altered neurohormonal response with important physiological consequences. Normally, the thyroid gland produces mainly T4 (a weaker receptor ligand) and, to a lesser extent, T3 (the biologically active form of TH). TH signaling is finely tuned within the tissue/cell in a precise time–space manner by three deiodinases and TRs [106]. DIO1 can both activate and inactivate THs and is expressed in the liver and kidneys, DIO2 catalyzes only phenolic ring deiodination and thus contributes to T3 production, while DIO3 is the physiological terminator of thyroid hormone activity and inactivates both T3 and T4 through tyrosyl ring deiodination [106]. T3 exerts its biological activity either by binding to thyroid nuclear receptors and regulating target gene expression (genomic action) or by binding to cytosolic partners and activating intracellular cascades (non-genomic action). THs and mainly T4 can also bind to the membrane αvβ3 integrin receptor, resulting in ERK activation [107]. The predominant mammalian TR isoforms include TRα1, TRβ1, TRβ2, TRβ3, TRβ4, and the TR variants that lack T3-binding capacity TRα2, TRα3, and TRαΔΕ6. In particular, TRα1 and TRβ1 are the best characterized mammalian thyroid receptors. TRα1 is most abundant in cardiac and skeletal muscle, bone, the gastrointestinal track, and the central nervous system. TRβ1 shows high expression in the liver, kidney and inner ear, while TRβ2 is predominant in the pituitary, cochlea, and hypothalamus [23]. Parallel to the hypothalamus–pituitary–adrenal (HPA) axis, the body exhibits stress-related changes in thyroid hormone metabolism (hypothalamus–pituitary–thyroid axis, HPT-axis) which result in low T3 levels in serum with normal T4 levels in less serious conditions and both low T3 and T4 serum levels in more severe cases. This response is known as the Non-thyroidal Illness Syndrome (NTIS, also known as low-T3 syndrome or euthyroid sick syndrome) [1] and seems to be an important determinant of survival in septic patients [108,109].

NTIS affects 60–70% of critically ill patients and is found nearly in 67% of sepsis patients. A meta-analysis of recent studies in patients with sepsis clearly pointed out that circulating T3 was lower in non-survivors versus survivors [108]. Furthermore, mortality in septic patients was found to be 13.4% in patients without thyroid hormone abnormalities, 50% in patients with low circulating T3, and rises up to 69.1% in patients with both low circulating T3 and T4 [109]. Furthermore, low T3 levels were associated with the risk of adverse cardiovascular events in adult patients with viral myocarditis [110].

These observations are in line with other acute pathological conditions, such as myocardial infarction [3], stroke [4], etc., indicating that changes in thyroid hormone metabolism in acute illness are not only indicators of disease severity but are common denominators in the pathophysiology of the stress response. Upon stress, there occurs an interaction of the adrenergic, hormonal (glucocorticoids), and immune systems over thyroid hormone metabolism and signaling in the body, which results in suppression of thyroid hormones [1,26,111,112,113]. In addition, in critically ill patients, a reduced activation (conversion of T4 to T3) and increased inactivation of TH occurs in tissues due to sepsis-induced tissue deiodinase changes (DIO1 and DIO3) [114,115]. Concomitant changes occur also in thyroid hormone nuclear receptors [112,113,115,116]. Furthermore, the HPT axis is deregulated (reviewed in detail in [26,111,113,117]). A critical-illness-related hypothalamic “insensitivity” to low circulating levels of T3 and T4 is responsible for lack of TSH-increase in NTIS [113]. Although the exact mechanisms of this remain unclear, evidence supports that hypothalamic TRH is suppressed by bacterial LPS [118] in the presence of low T3/T4 hypothalamic levels [119]. However, this response is not unique in bacterial sepsis but also applies for viral sepsis, such as SARS-CoV-2 infection, indicating that other mechanisms beyond LPS are involved [120,121]. Eventually, NTIS upon stress (sepsis or any other acute pathological condition) seems to be of physiological importance–albeit of unknown precise significance–as TH regulates several pathways involved in cell differentiation, growth, apoptosis, metabolism, and mitochondrial biogenesis [27,112,113].

## 6. TH-Effects in Tissue Hypoxia Induced by Sepsis

### 6.1. TH and Tissue Hypoxia in Sepsis

We recently showed a novel action of TH on tissue hypoxia in experimental sepsis, as previously observed in myocardial infarction [122,123]. Interestingly, analogies exist between sepsis and acute myocardial infarction and the presence of microvascular impairment was associated with worse outcome of both conditions [124]. Tissue hypoxia occurs early after sepsis, with increase in lactate levels. We recently showed that in an experimental model of sepsis in mice (CLP-induced sepsis), tissue hypoxia occurred in the heart and liver in the early phase of sepsis before hemodynamic deterioration occurred [122]. We detected tissue hypoxia by pimonidazole staining, an immunohistochemical technique able to detect tissue pO2  <  10 mmHg. Furthermore, we detected early increased levels of circulating lactate in these mice as a result of early tissue hypoxia and/or metabolic switch towards glycolysis [122]. As discussed above, tissue oxygen below 10 mmHg results in activation of HIF1α-dependent regulatory mechanisms that promote pathologic angiogenesis, changes in immune response, and determine sepsis-induced injury progression [87]. Importantly, early T3 treatment prevented tissue hypoxia and significantly reduced circulating lactate. Here, T3 treatment significantly reduced the extent of myocardial tissue hypoxia from 4 ± 0.5% in untreated animals to 1.5% ± 0.5 in treated ones, at 18 h post sepsis initiation [122]. In accordance, T3 treatment also reduced liver tissue hypoxia [122].

Still, the underlying mechanisms of this novel favorable effect of T3 on sepsis-induced tissue hypoxia remain to be elucidated.

### 6.2. TH and Mitochondrial Function

Sepsis has profound effects on cellular energy and is associated with a defect in mitochondrial functions and biogenesis (for a detailed review see [125]). TH signaling seems to be involved in this response. Experimental sepsis in mice resulted in reduced T4 serum levels and liver Dio1 (Type I iodothyronine deiodinase) expression, hallmarks of NTIS. These mice also displayed down-regulated TRα, TRβ, soluble-carrier (Slc) family members Slc16a10, and Slc16a2 expression in the diaphragm, suggesting that sepsis impairs TH signaling at the tissue level as well [126]. Similar data were also confirmed for the liver after sepsis [116]. The expression of Ppargc1a (encoding peroxisome proliferator-activated receptor gamma coactivator 1-alpha, PGC1a, a key regulator of energy metabolism [127]) was also down-regulated, which correlated with a decrease in the number of total mitochondria, increase in the percentage of injured mitochondria, and down-regulation of respiratory chain complex 2 and 3 mRNA expression, collectively resulting in reduced oxidative phosphorylation capacity. However, TH treatment ameliorated the impact of sepsis on the total number and percentage of healthy mitochondria and improved expression of complex II and III, which were down-regulated by sepsis [126].

### 6.3. TH and Hemorheology

Red cell blood rheology appears to be altered in sepsis, with important physiological consequences [80]. Changes in red blood cell deformation and increased red cell aggregation can impair microcirculation and lead to tissue hypoxia and injury.

The Erythrocyte Sedimentation Rate (ESR) is, among others, a common clinical tool that indicates erythrocytes’ aggregation when increased. Sepsis increases ESR non-specifically. Apparently, changes in ESR are linked to the function of microcirculation [128]. The flow of blood in microvessels essentially depends on viscous forces (i.e., the viscosity of blood) due to low flow velocities. Accordingly, when the attractive forces between erythrocytes are increased (represented by an increased ESR and thus aggregation of erythrocytes) and are greater than the sheer force produced by microvascular flow, tissue perfusion itself cannot be sustained, leading to loss of capillary perfusion. Thus, any reduction in such an increased ESR indicates improved blood viscosity and better microvascular flow [128].

The potential role of TH on erythrocytes’ aggregation and rheology in sepsis remains largely unknown. Surprisingly, preliminary data from a small investigator-driven clinical study, applying T3 in septic patients with COVID19 infection, showed that acute administration of T3 significantly reduced ESR within 48 h of infusion, in a concentration-dependent manner [129]. Interestingly, this effect was associated with a trend towards lower troponin levels in treated patients [129]. This novel T3 action could be relevant to the observed favorable effect on sepsis-induced microcirculatory failure and tissue hypoxia [122] and merits further investigation (Figure 1).

### 6.4. TH Effects on Relevant for Sepsis Immune System Functions

TH is critical for immune system function. Chemotaxis, phagocytosis, generation of reactive oxygen species, and cytokine synthesis and release, are altered in hypo- and hyper-thyroid conditions, even though for many immune cells no clear correlation has been found thus far [117,130]. On the other hand, both infection and inflammation affect TH-levels through the hypothalamus–pituitary–thyroid gland axis and, as a direct consequence, THs mechanisms of action [117].

Even small differences within the normal range of T3 and T4 levels are associated with changes in markers of inflammation and immunity [131]. In healthy individuals, a higher ratio of T3/T4 is shown to be associated with increased monocyte phagocytic activity, higher IL-2 receptor density on CD3+ T-lymphocytes, and higher absolute counts of natural-killer T-cells (NKTs) [132]. Thyroid hormone *treatment* modulates T-helper (Th1/Th2) lymphocyte *responses* and thereby amplifies host defenses against viral infections [133]. Along this line, ligand-bound TRα represses the activation of NF-κB (nuclear factor kappa B) involved in polarizing macrophages and inhibits nuclear localization of p65 in macrophages, through stabilization of MAPK-phosphatase 1 (MKP1) via MAPK (MKP1), thereby leading to down-regulation of proinflammatory cytokines [134]. TRa is also found upregulated in macrophages and brain microglia in the subacute, inflammatory active, phase of stroke [135,136], suggesting a possible physiological role for the receptor under different TH local-tissue concentrations. On the other hand, TRα-deficient irradiated chimeric mice exhibited exacerbated kidney injury in the unilateral ureteral obstructed kidney. Macrophages isolated from the obstructed kidneys of mice solely lacking TRα in the bone marrow displayed increased expression of proinflammatory cytokines, including IL-1β, compared with wild-type mice. Ligand-bound TRα on macrophages protected the kidney, again via inhibiting the NF-κB pathway, by fine regulation of the pro- and anti-inflammatory balance that controls the development of chronic kidney disease [134]. Furthermore, ligand-bound TRα in macrophages down-regulated LPS-induced IL-1β expression, which is a target of the NF-κB pathway, during acute and chronic inflammation of the liver [137].

TH stimulation also has profound effects on dendritic cells (DC) phenotype. Incubation of human peripheral blood mononuclear cells with TH enhances their ability to differentiate into functional DCs [138]. Stimulation of murine bone marrow-derived DCs with physiological levels of T3 can result in the initiation of the adaptive immune response by induction of DC maturation, increased IL-12 production, improved antigen cross presentation, and enhanced DC ability to stimulate a cytotoxic T-cell response and trigger antigen-specific responses in vivo [139,140,141]. Cell survival and ability to migrate to lymph nodes in vivo are also enhanced [139]. The effects of T3 stimulation are mediated via the TRβ1 receptor and the Akt and NFκB pathways independently of the PI3K (phosphatidylinositol 3-kinase) pathway [140,141]. The promoter region of the TRβ1 gene contains an NFκB response element, which upregulates TRβ1 expression after T3 stimulation, suggesting a regulatory feedback loop [141]. The immune cascade in the paradigm of sepsis is highly complex (a dynamic balance of pro- and anti-inflammatory pathways) and TH can exert multiple actions at different levels. Further studies are needed to elucidate these effects (Figure 1).

## 7. Clinical Safety and Feasibility of TH-Treatment in Critically Ill Patients

Until now, the extensive use of T3 in clinical practice has been limited due to a long-standing belief that T3 may aggravate ischemia in hypoxic conditions such as myocardial infarction, bypass surgery, or sepsis. Based on this belief, T3 may increase metabolic rate and oxygen consumption, resulting in detrimental effects during hypoxia. Nevertheless, several studies have proven the opposite, namely, that T3 administration in clinical settings of hypoxia is safe, even in severely ill patients, as reviewed before [142] and discussed below. Current recommendations for NTIS treatment in critically ill patients are based on a few, rather small, clinical trials, are inconclusive, and suggest only a symptomatic treatment of the overt hypothyroidism within the NTIS spectrum [142].

T3 has been safely used as an inotrope to support impaired hemodynamics in adults and even in children with severe low-cardiac-output syndrome in whom conventional treatment had failed [143]. In children undergoing cardiac surgery for congenital heart defects, circulatory arrest induced by deep hypothermia results in increase in TSH and reduced T3 [144]. Preoperative oral TH therapy in these children was shown to protect against myocardial I/R injury by increasing HSP70 [145]. In accordance, the low T3 levels of NTIS after successful resuscitation and therapeutic hypothermia, have been associated with death or severe coma [146]. The efficacy of T3 on cardiac hemodynamics and the side effects have been also tested in several by-pass trials in adults. A meta-analysis showed that TH may have beneficial effect on cardiac hemodynamics without severe side effects [147]. Both high- and low-dose intravenous (iv) T3 treatment resulted in increased cardiac index after coronary artery bypass graft surgery (CABG) with no adverse effect of high T3-dose on mortality. Similarly, a randomized double-bind placebo–control trial on reperfusion-injury after CABG with 440 patients showed a significantly increased cardiac index with lower troponin release (as a cardiac injury index), after T3 treatment versus placebo and glucose–insulin–potassium treatment [148]. Another randomized double-blind controlled prospective study on patients undergoing CABG also showed that perioperatively administered T3 significantly reduces the mean norepinephrine use in the first six hours after removal of an aortic cross clump and lowers the incidence of atrial fibrillation after cardiac operations [149]. TH therapy for supporting donor heart hemodynamics has extensively been used in cardiac transplantation, indicating that TH may protect the donor heart against hypoxic injury [150]. This effect was also confirmed from the analysis of 66,629 cardiac donors where T3/T4 treatment was associated with procurement of significantly greater numbers of hearts and improved post-transplantation graft survival [151,152]. In alignment with preclinical experimental data (where T3 improves post-ischemic cardiac function, while limiting apoptosis [19]), the efficacy and safety of the use of triiodothyronine has been investigated in patients with anterior STEMI (ST-segment elevation myocardial infarction) undergoing angioplasty (ThyRepair trial EudraCT: 2016-000631-40). This study has been successfully completed without major safety issues [124]. A small and transient increase in heart rate was only observed in patients during high-dose T3 treatment [124].

Evidently, emerging but still few data suggest that T3 supplementation in non-cardiac critically ill patients can be feasible and safe. A retrospective observational study in such patients indicated that T3 supplementation was safe and reduced new cardiac-related adverse events compared to baseline [153]. In an old, preliminary, single-arm study [154], with 11 patients in severe septic shock that were supported hemodynamically with dopamine, T3 was administered by continuous intravenous infusion in doses of 100–200 μg/24 h for up to 28 days and resulted in increased arterial blood pressure, termination of dopamine dependence, and lack of significant adverse events [154]. More recently, our preliminary data from a phase II randomized double-blind placebo–controlled study to demonstrate the safety and efficacy of T3 in critically ill COVID-19 patients (Thy-Support study, NCT04348513, EudraCT: 2020-001623-13) showed no major side effects even in this severely septic population [129].

Sepsis remains a clinical condition which carries high morbidity and mortality when it evolves to septic shock with impaired hemodynamics and organ hypoperfusion and failure. The current treatments aim to control infection and inflammation and support (macro-)hemodynamics. However, there is no available treatment to prevent early tissue hypoxia that occurs at a time where hemodynamics are preserved and induces tissue remodeling resulting in septic shock and MOF. The novel action of T3 to prevent microcirculatory failure and tissue hypoxia in experimental sepsis should be further therapeutically exploited (patents apply).

## 8. Conclusions

A growing body of experimental and clinical evidence reveals an evolutionary conserved action of thyroid hormone (TH) to adapt tissue to hypoxic conditions. In addition, TH has favorable effects on microcirculatory, mitochondrial, and immune system function. These actions of T3 could be therapeutically exploited in the setting of critical illness, such as sepsis, in which tissue hypoxia prevails.

## 9. Patents

The following patents are relevant to the work in this manuscript.

PCT/EP2019/087056. L-triiodothyronine (T3) for use in limiting microvascular obstruction.

Greek Patent Office, case number: 22-0002577373. Composition comprising L-triiodothyronine (T3) for use in the treatment of critically ill patients with coronavirus infection.

PCT/4972/2021. Pharmaceutical composition comprising L-triiodothyronine (T3) for use in the treatment of tissue hypoxia and sepsis (pending).

## Figures and Tables

**Figure 1 jcm-10-05855-f001:**
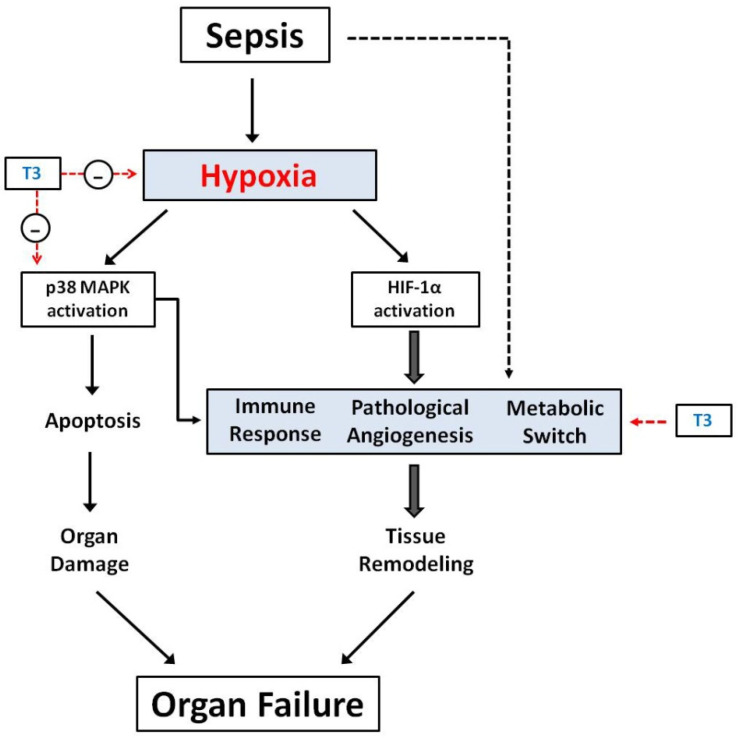
Tissue hypoxia has a critical role in the main mechanisms that results in organ failure due to sepsis (the dotted black line represents the hypoxia-independent complex pathophysiological mechanisms, resulting in sepsis immune response, pathological angiogenesis, and metabolic switch; these are not reviewed here). Here, we depict possible sites of TH action (red dotted lines): (1) triiodothyronine (T3) improves tissue hypoxia and inhibits pro-apoptotic kinase signaling activation and (2) affects the other septic mechanisms in a complex way. Eventually, T3 may prevent organ failure.

**Table 1 jcm-10-05855-t001:** Accumulating experimental evidence shows that thyroid hormone plays a critical role in protection/adaptation of the heart to hypoxic/ischemic injury. CAL = Coronary Artery Ligation.

Study	Type of Treatment	Model	Outcome
Pantos et al., 2002 [15]	Pre-treatment with T4	Isolated rat heart	Increased recovery of function
Kuzman et al., 2005 [24]	Pre-treatment with T3	Neonatal rat cardiomyocytes	Increased cell viability, reduced apoptosis
Pantos et al., 2011 [18]	Treatment with T3 post-ischemia	Isolated rat heart	Increased recovery of function, reduced injury and apoptosis
Pantos et al., 2009 [19]	Treatment with T3 post-ischemia	Isolated rat heart	Increased recovery of function, reduced injury
Chen et al., 2008 [25]	Treatment with T3 after infarction	In vivo CAL rat heart	Improved LV function, reduced apoptosis
Forini et al., 2014 [21]	Treatment with T3 after infarction	In vivo CAL rat heart	Reduced infarct size and mitochondrial impairment
Fang et al., 2019 [20]	Pre-treatment with T3	Isolated rat heart	Improved LV function
Karakus et al., 2021 [22]	Pre-treatment with T3 polymeric nanoparticles	Neonatal rat cardiomyocytes	Improved hypoxic cell damage

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
