# Peer review of "Effects of Thyroid Hormone on Tissue Hypoxia: Relevance to Sepsis Therapy"

_jcm, 2021, doi:10.3390/jcm10245855_

Round 1
Reviewer 1 Report
The issue is very interesting in the context of many controversions connected with the treatment with thyroid hormones, especially T3 in severe conditions.
However, the manuscript needs major revision.
The authors declared, that “in this critical review, we present evidence related to the potential effects of TH on microcirculatory failure and tissue hypoxia, which are key players in the development of tissue injury, remodeling and organ failure in the paradigm of sepsis”.
But the first 3 parts of the paper are not strongly related to the declared aim of the study. I propose to focus the attention on the interconnectedness between hypoxia and thyroid hormones. Maybe it would be worthy to explain the changes in hypoxia in patients with overt hypothyroidism as well as in opposite condition – with thyrotoxicosis.
In line 205: Upon stress, an interaction between thyroid hormone signaling, the adrenergic system and immune system occurs and results in suppression of thyroid hormones [58,59].
This citations do not explain this relations, especially the mechanism of suppression of thyroid hormones, it should be supported by source citations, another way the sentence sounds as only speculation. It should be more precise.
Line 208- in the sentence:
“changes in the degradation of thyroid hormones in tissue and in thyroid hormone nuclear receptors”- this should be also illustrated by source citations.
In the paper there are many not explained abbreviations
i.e. : Pdgf in line 316
in the line 281, 282 and 282: CLP,
the same in line 358: Dio , Slc16a
In the part 5.1 in the line 251-254 authors used different symbols of Thyroid hormone receptors than in the part 5.6, it should be unified in the whole text
The same with “signaling” and “signalling” .
The two similar sentences starting in line 342 are incomprehensible, what authors want to show?
“This novel T3 action could be relevant to the observed favorable effect on sepsis-induced microcirculatory failure and tissue hypoxia [77] and merits further investigation. Figure 3.
Although this novel action of T3 points to a favorable action of T3 on sepsis-induced microcirculatory failure and tissue hypoxia and merits further investigation. Figure 1.” – it is difficult to understand why the repeat?
Authors wrote in the line 379, that :
“Until now, the extensive use of T3 in clinical practice has been limited due to long standing belief that T3 may aggravate ischemia in hypoxic conditions such as myocardial infarction, bypass surgery or sepsis. Nevertheless, several studies have proven that T3 (…) is safe, even in severely ill patients.
It needs the citations of studies supporting this data (each of the study).
Part 6 is to short, when authors want to translate the molecular basis directly to the clinical use. It should be illustrated with the results in human with and without treatment. In this part I suggest also include the data from patients with low T3 in hypothermia, it is worthy to present and discuss the current recommendation of the use of thyroid hormones in severe conditions and NTIS.
The conclusions are not fully supported by previous parts of the manuscript, especially the first sentence is unauthorized.
Author Response
The issue is very interesting in the context of many controversies connected with the treatment with thyroid hormones, especially T3 in severe conditions.
However, the manuscript needs major revision.
- The authors declared, that “in this critical review, we present evidence related to the potential effects of TH on microcirculatory failure and tissue hypoxia, which are key players in the development of tissue injury, remodeling and organ failure in the paradigm of sepsis”.
But the first 3 parts of the paper are not strongly related to the declared aim of the study. I propose to focus the attention on the interconnectedness between hypoxia and thyroid hormones.
Reply to Reviewer: we have substantially revised, restructured and complemented the entire manuscript with new evidence to enhance the effects and interconnection of thyroid hormones (TH) on tissue hypoxia. The first parts now focus with detailed evidence on the TH-hypoxia interconnection in different tissues (heart, liver, brain, etc). Following these, we then prove that sepsis is also a tissue hypoxia problem and, as such, TH could play a significant role for its treatment.
- Maybe it would be worthy to explain the changes in hypoxia in patients with overt hypothyroidism as well as in opposite condition – with thyrotoxicosis.
Reply to Reviewer: We thank the reviewer for this suggestion. Since data from hyperthyroid or hypothyroid patients are very few, we have included substantial evidence showing that THs levels regulate the HIF-1a and the hypoxia response pathway. Furthermore, we included important preclinical data showing that antecedent changes in thyroid hormone levels significantly affect the response of various tissues to hypoxia/ischemia. Section 2 “THs and tissue hypoxia “, table 1 &2.
- In line 205: "Upon stress, an interaction between thyroid hormone signaling, the adrenergic system and immune system occurs and results in suppression of thyroid hormones [58,59]." This citations do not explain this relations, especially the mechanism of suppression of thyroid hormones, it should be supported by source citations, another way the sentence sounds as only speculation. It should be more precise.
Line 208- in the sentence: “changes in the degradation of thyroid hormones in tissue and in thyroid hormone nuclear receptors”- this should be also illustrated by source citations.
Reply to Reviewer: we thank the reviewer for the notices. We have corrected and updated the relevant lines and sentences. Page 10, lines 11-16
- In the paper there are many not explained abbreviations. i.e. : Pdgf in line 316; in the line 281, 282 and 282: CLP; the same in line 358: Dio, Slc16a. In the part 5.1 in the line 251-254 authors used different symbols of Thyroid hormone receptors than in the part 5.6, it should be unified in the whole text. The same with “signaling” and “signalling”.
Reply to Reviewer: thank you for your notifications, we have corrected all abbreviations and inconsistencies as suggested.
- The two similar sentences starting in line 342 are incomprehensible, what authors want to show? “This novel T3 action could be relevant to the observed favorable effect on sepsis-induced microcirculatory failure and tissue hypoxia [77] and merits further investigation. Figure 3."
"Although this novel action of T3 points to a favorable action of T3 on sepsis-induced microcirculatory failure and tissue hypoxia and merits further investigation. Figure 1.” – it is difficult to understand why the repeat?"
Reply to Reviewer: thank you for pointing this typing error out, we have corrected it.
- Authors wrote in the line 379, that: “Until now, the extensive use of T3 in clinical practice has been limited due to long standing belief that T3 may aggravate ischemia in hypoxic conditions such as myocardial infarction, bypass surgery or sepsis. Nevertheless, several studies have proven that T3 (…) is safe, even in severely ill patients." It needs the citations of studies supporting this data (each of the study).
Reply to Reviewer: we have extensively revised and updated the relevant section to include -to our knowledge- the majority of clinical studies in different set-ups and populations that show a safe T3 administration in severely ill humans. Citations supporting this data are provided next to each described study in the revised section.
- Part 6 is to short, when authors want to translate the molecular basis directly to the clinical use. It should be illustrated with the results in human with and without treatment. In this part I suggest also include the data from patients with low T3 in hypothermia, it is worthy to present and discuss the current recommendation of the use of thyroid hormones in severe conditions and NTIS.
Reply to Reviewer: we have extensively revised and expanded the respective part, to illustrate the results from clinical studies with and without T3 treatment and usage of THs as suggested. Data from patients with low T3 in hypothermia have been discussed and current recommendations for NTIS in critically ill have been added.
- The conclusions are not fully supported by previous parts of the manuscript, especially the first sentence is unauthorized.
Reply to Reviewer: we have completely revised the conclusion of our manuscript as suggested.
Reviewer 2 Report
Very well-written and interesting review. The introduction lacks information directly related to the main topic of the article - thyroid hormones effects on tissue hypoxia. Please check if the abbreviations are correctly described when they first appear in the text (eg HT line 42). The role of T3 in my opinion is not shown clearly on Figure 1, especially T3 effect on immune response, pathological angiogenesis and metabolic switch - T3 acts on all of them or only on metabolic switch? Figure 2 was presented in Mourouzis, I.S., Lourbopoulos, A.I., Trikas, A.G. et al. Triiodothyronine prevents tissue hypoxia in experimental sepsis: potential therapeutic implications. ICMx 9, 17 (2021). https://doi.org/10.1186/s40635-021-00382-y - in my opinion it is not necessary to present this figure once again in this review. Figure 3 a and b was presented in Pantos C, Apostolaki V, Kokkinos L, Trikas A, Mourouzis I. Acute triiodothyronine treatment and red blood cell sedimentation rate (ESR) in critically ill COVID-19 patients: A novel association? Clin Hemorheol Microcirc. 2021 Jun 11. doi: 10.3233/CH-211215. Epub ahead of print. PMID: 34151781 - is it necessary to present this information once again in the same form in this review?
Author Response
Very well-written and interesting review.
- The introduction lacks information directly related to the main topic of the article - thyroid hormones effects on tissue hypoxia.
Reply to Reviewer: we have substantially revised, restructured and complemented the entire manuscript with new evidence to enhance the effects and interconnection of thyroid hormones (TH) on tissue hypoxia, as suggested by both reviewers.
- Please check if the abbreviations are correctly described when they first appear in the text (eg HT line 42).
Reply to Reviewer: thank you for your notifications, we have corrected all abbreviations and inconsistencies as suggested.
- The role of T3 in my opinion is not shown clearly on Figure 1, especially T3 effect on immune response, pathological angiogenesis and metabolic switch - T3 acts on all of them or only on metabolic switch?
Reply to Reviewer: the role of T3 on immune response, pathological angiogenesis and metabolic switch is complex. In the present Figure we had aimed to present a "rough direction" regarding the sites of T3 actions, namely to point that T3 (according to all currently available data presented in our manuscript) may have effects on 1) tissue hypoxia, 2) p38MAPK activation and 3) all downstream or parallel mechanisms of sepsis, i.e. immune response, pathological angiogenesis and metabolic switch. We have now restructured Figure 1 to keep its "rough directive" concept and clarify that T3 can act on all 3 "immune response, pathological angiogenesis and metabolic switch".
- Figure 2 was presented in Mourouzis, I.S., Lourbopoulos, A.I., Trikas, A.G. et al. Triiodothyronine prevents tissue hypoxia in experimental sepsis: potential therapeutic implications. ICMx 9, 17 (2021). https://doi.org/10.1186/s40635-021-00382-y - in my opinion it is not necessary to present this figure once again in this review.Figure 3 a and b was presented in Pantos C, Apostolaki V, Kokkinos L, Trikas A, Mourouzis I. Acute triiodothyronine treatment and red blood cell sedimentation rate (ESR) in critically ill COVID-19 patients: A novel association? Clin Hemorheol Microcirc. 2021 Jun 11. doi: 10.3233/CH-211215. Epub ahead of print. PMID: 34151781 - is it necessary to present this information once again in the same form in this review?
Reply to Reviewer: we have removed both Figures as suggested.
Reviewer 3 Report
Sepsis is an extreme response triggered by an infection that spreads through the entire body and can rapidly lead to organ failure, tissue damage and, eventually, death. One of the main pathophysiological mechanisms of septic shock is microcirculatory failure followed by tissue hypoxia, that can lead to tissue-reprogramming programs that aggravate the condition.
In this review, the authors extensively review the clinical implications of myocardial infarction and microvasculature accidents secondary to sepsis in the prognosis of patients, as well as some of the molecular mechanisms that can be involved in these processes, such as Hif⍺ or p38 MAPK pathways. They also review the involvement of thyroid hormones, specifically T3, in the regulation of the immune system response to sepsis via inhibition of NFkb and a fine regulation of pro/anti-inflammatory cytokine balance, as well as its relationship with hypoxia and tissue remodeling. This is an interesting and well-organized review but as this is a broad topic, some of the statements of the manuscript should be clarified.
Majors:
-The metabolism (activation and degradation) of thyroid hormones by deiodinases as well as the main mechanism of thyroid hormones action (modulation of gene expression and extragenomic actions) should be briefly explained in the section “Sepsis and neurohormonal response” to better understand the pathophysiology related to thyroid hormones in sepsis.
-In section 3.1, the authors describe the potential effects of p38 MAPK and stress on cell proliferation / differentiation and tissue remodeling, but they don’t deepen into what effect p38 MAPK has or by which pathways this protein triggers these responses.
-In Figure 1 it should be illustrated and clarified if only T3, or both T3 and T4 inhibits hypoxia. The possible effect of T4 on the different issues assessed in the review is hardly mentioned and could be better explained. This figure should be nearest section 4.
-In section 4, the authors refer to the effect of stress as an interactor between thyroid hormones, immune and adrenergic system that causes thyroid hormones degradation in tissues. They also mention an effect on T3 receptors, but they don’t give any specific information about the pathophysiological mechanisms related to this effect. Please mention them, or in the case that they are not characterized please specified it.
-In section 4, regarding the NTIS explanation, the effects of stress and thyroid hormones in bacteriological septic mode (LPS injections) are discussed. Please clarify if this apply to viral sepsis or it is specific of bacterial sepsis. In addition, it would be very informative to know data from other cases of septic shocks.
- In section 5.4 authors don’t discuss about the extragenomic effects of T4 in angiogenesis and pro-angiogenic factor gene expression; which is proven to be the most relevant action of thyroid hormones in blood-vessel formation and remodeling. Also, they only refer to the direct genomic action of T3, never mentioning the effects of T4, when extragenomic action in angiogenesis regulation is more noticeable than T3.
-In section 5.6, authors talk about DIO1 being a hallmark of NTIS, but when the authors first describe NTIS and its hallmarks they don’t say anything about deiodinases, neither they do when they explain thyroid hormones metabolism in point 4.
-In section 6, must be discussed why it was long believed that T3 aggravated ischemia in hypoxia and myocardial infarction, why it is now proven that it doesn’t do it and why this review is important to understand the differences between the past convictions and the present knowledge about this issue.
-Asterisks in Figure 2d do not coincide with the published figure in PMID: 33834320, please check and correct and also mark with a different letter every photo/graphic.
Minors:
-Please clarify the abbreviation TH and use it properly along the manuscript. TH is usually used as an abbreviation for both T3 and T4. In the manuscript the abbreviation TH sometimes can be understood as T3 and T4 and other times as T3, and this is confusing.
-There are writing errors, such as a lack of space between many words. Sometimes there are three words together. This should be revised along the manuscript.
-Please revise all the abbreviations: cite them the first time they appear, and use them afterwards. In addition, there is a lack of uniformity in the name of some of the abbreviations as for example p38 MAPK, being written with and without a space between words indifferently. Oxidative phosphorylation complexes II and III are referred both as II/III and ⅔ in section 5.6. In addition, some of the abbreviations are not explained such as vRNP and TRs.
-Line 74 please add “to” before “several”
-As a suggestion, the molecular mechanisms by which thyroid hormones can mediate their effects on tissue remodeling, metabolic switching and cell redifferentiation should be better described whenever possible.
Author Response
Majors:
-The metabolism (activation and degradation) of thyroid hormones by deiodinases as well as the main mechanism of thyroid hormones action (modulation of gene expression and extragenomic actions) should be briefly explained in the section “Sepsis and neurohormonal response” to better understand the pathophysiology related to thyroid hormones in sepsis.
Reply to Reviewer: The metabolism (activation and degradation) of thyroid hormones by deiodinases as well as the main mechanism of thyroid hormones action (modulation of gene expression and extragenomic actions) are briefly explained in the section “Sepsis and neurohormonal response”. We added the following text:
The thyroid gland produces mainly T4 (a weaker receptor ligand) and to a lesser extent T3 (the biologically active form of TH). TH signalling is finely tuned within tissue/cell in a precise time–space window by three deiodinases and TRs. DIO1 can both activate and inactivate THs and is expressed in the liver and kidneys, DIO2 catalyses only phenolic ring deiodination and thus contributes to T3 production while DIO3 is the physiological terminator of thyroid hormone activity and inactivates both T3 and T4 through tyrosil ring deiodination. T3 exerts its biological activity either by binding to thyroid nuclear receptors and regulating target gene expression (genomic action) or by binding to cytosolic partners (and activating intracellular cascades (non-genomic action). THs and mainly T4 can also bind to the membrane αvβ3 intergrin receptor resulting in ERK activation. The predominant mammalian TR isoforms include TRα1, TRβ1, TRβ2, TRβ3, TRβ4 and the TR variants that lack T3-binding capacity TRα2, TRα3 and TRαΔΕ6. Especially TRα1 and TRβ1 are the best characterized mammalian thyroid receptors. TRα1 is most abundant in cardiac and skeletal muscle, bone, gastrointestinal track and central nervous system. TRβ1 shows high expression in the liver, kidney and inner ear, while TRβ2 is predominant in the pituitary, cochlea and hypothalamus.
Luongo C, Dentice M, Salvatore D. Deiodinases and their intricate role in thyroid hormone homeostasis. Nat Rev Endocrinol. 2019 Aug;15(8):479-488. doi: 10.1038/s41574-019-0218-2
-In section 3.1, the authors describe the potential effects of p38 MAPK and stress on cell proliferation / differentiation and tissue remodeling, but they don’t deepen into what effect p38 MAPK has or by which pathways this protein triggers these responses.
Reply to Reviewer: In section entitled ‘The molecular basis of tissue hypoxia in sepsis: HIF-1a and p38 MAPK’ we added relevant information briefly describing the pathway by which p38 MAPK triggers these responses. We added the following text:
p38 MAPK mediates its actions via phosphorylation of a broad range of substrates (reviewed in detail in [96]). In brief, phosphorylation of MSK (mitogen- and stress-activated kinase) 1 and 2 can regulate transcription factors such as c-AMP Response Element-Binding Protein (CREB), c-AMP-dependent transcription factor (ATF-1), the NF-κB isoform p65, signal transducer and activator of transcription (STAT 1 and 3) and p53. Notably, p38 MAPK can phosphorylate Hsp27 (heat-shock protein 27) mediating actin filament remodelling. A large number of cytosolic proteins important for cell proliferation and survival/apoptosis are shown to be regulated by p38 MAPKs, including phospholipase A2, the tau protein, cyclin D1, and Bcl-2 family proteins[96]. Another example of p38 MAPK substrate is the FGFR1 (fibroblast growth factor receptor 1) which is translocated from the extracellular space into the cytosol and nucleus regulating cell growth. Furthermore, p38 MAPK is directly implicated in transcription of several genes involved in the inflammatory response, such as IL-6, IL-8, IL-12p40 and MCP-1 (monocyte chemoattractant protein 1) via histone H3 phosphorylation[96].
-In Figure 1 it should be illustrated and clarified if only T3, or both T3 and T4 inhibits hypoxia. The possible effect of T4 on the different issues assessed in the review is hardly mentioned and could be better explained. This figure should be nearest section 4.
Reply to Reviewer:Existing evidence suggests that T3 inhibits hypoxia. Data about an effect of T4 or not on sepsis induced tissue hypoxia do not exist so far. Thus, we confine the figure based on the available evidence. We tried to assess the effect of T4 on the different issues assessed in the review whenever possible.
-In section 4, the authors refer to the effect of stress as an interactor between thyroid hormones, immune and adrenergic system that causes thyroid hormones degradation in tissues. They also mention an effect on T3 receptors, but they don’t give any specific information about the pathophysiological mechanisms related to this effect. Please mention them, or in the case that they are not characterized please specified it.
Reply to Reviewer:The relevant section entitled Sepsis and Neurohormonal response has been extensively revised based on reviewer suggestions.
-In section 4, regarding the NTIS explanation, the effects of stress and thyroid hormones in bacteriological septic mode (LPS injections) are discussed. Please clarify if this apply to viral sepsis or it is specific of bacterial sepsis. In addition, it would be very informative to know data from other cases of septic shocks.
Reply to Reviewer:We thank the reviewer for this comment. The interaction of stress and thyroid hormones resulting in NTIS applies to both bacterial and viral infections. We clarified this issue in the text.
- In section 5.4 authors don’t discuss about the extragenomic effects of T4 in angiogenesis and pro-angiogenic factor gene expression; which is proven to be the most relevant action of thyroid hormones in blood-vessel formation and remodeling. Also, they only refer to the direct genomic action of T3, never mentioning the effects of T4, when extragenomic action in angiogenesis regulation is more noticeable than T3.
Reply to Reviewer:We thank the reviewer for this comment. We now extended the relevant section ‘TH and angiogenesis’ including the important extragenomic actions of T4
-In section 5.6, authors talk about DIO1 being a hallmark of NTIS, but when the authors first describe NTIS and its hallmarks they don’t say anything about deiodinases, neither they do when they explain thyroid hormones metabolism in point 4.
Reply to Reviewer: The role of deiodinases in NTIS has now been described in section 5. Information about the role of deiodinases in TH metabolism has also been added.
-In section 6, must be discussed why it was long believed that T3 aggravated ischemia in hypoxia and myocardial infarction, why it is now proven that it doesn’t do it and why this review is important to understand the differences between the past convictions and the present knowledge about this issue.
Reply to Reviewer:Section 6 has now been extensively revised to address these issues according to reviewer suggestion.
-Asterisks in Figure 2d do not coincide with the published figure in PMID: 33834320, please check and correct and also mark with a different letter every photo/graphic.
Reply to Reviewer:Figure 2 has been excluded according to suggestion from reviewer 2
Minors:
-Please clarify the abbreviation TH and use it properly along the manuscript. TH is usually used as an abbreviation for both T3 and T4. In the manuscript the abbreviation TH sometimes can be understood as T3 and T4 and other times as T3, and this is confusing.
Reply to Reviewer:The abbreviation TH stands for both T4 and T3. We tried to use it in an appropriate manner along the manuscript.
-There are writing errors, such as a lack of space between many words. Sometimes there are three words together. This should be revised along the manuscript.
Reply to Reviewer:writing errors were corrected along the manuscript
-Please revise all the abbreviations: cite them the first time they appear, and use them afterwards. In addition, there is a lack of uniformity in the name of some of the abbreviations as for example p38 MAPK, being written with and without a space between words indifferently. Oxidative phosphorylation complexes II and III are referred both as II/III and ⅔ in section 5.6. In addition, some of the abbreviations are not explained such as vRNP and TRs.
Reply to Reviewer:All abbreviations have been revised according to reviewer suggestions
-Line 74 please add “to” before “several”
Reply to Reviewer:“to” has been added, Page 7, line 10
-As a suggestion, the molecular mechanisms by which thyroid hormones can mediate their effects on tissue remodeling, metabolic switching and cell redifferentiation should be better described whenever possible.
Reply to Reviewer:Mechanisms of action of thyroid hormones on tissue remodeling, metabolic switching and cell redifferentiation were better described whenever possible.
Round 2
Reviewer 1 Report
The revision was well done and improved the value of the paper.